# Antibiotic prescription patterns in patients with suspected urinary tract infections in Ecuador

**Xavier Sánchez**[1,2]ʘ¤, **Alicia Latacunga**[3]‡, **Iván Cárdenas**[3]‡, **Ruth Jimbo-Sotomayor**[1,2]ʘ*, **Santiago Escalante**[1]‡

**1** Centro de Investigación Para la Salud en América Latina (CISeAL), Facultad de Medicina, Pontificia Universidad Católica del Ecuador (PUCE), Quito, Ecuador, **2** Community and Primary Care Research Group – Ecuador (CPCRG-E), Quito, Ecuador, **3** Postgrado de Medicina Familiar y Comunitaria, Pontificia Universidad Católica del Ecuador (PUCE), Quito, Ecuador

ʘ These authors contributed equally to this work.
¤ Current address: Centro de Investigación para la Salud en América Latina (CISeAL), Pontificia Universidad Católica del Ecuador (PUCE), Quito, Ecuador
‡ AL, IC and SE also contributed equally to this work.
* rejimbo@puce.edu.ec

**Data Availability Statement:** files are available from the Mendeley Data, as public database: Cite: Sánchez, Xavier (2023), "Antibiotic Prescription Patterns in Patients with Suspected Urinary Tract

## Abstract

### Background

Urinary tract infections (UTI) are among the most common cause to prescribe antibiotics in primary care. Diagnosis is based on the presence of clinical symptoms in combination with the results of laboratory tests. Antibiotic therapy is the primary approach to the treatment of UTIs; however, some studies indicate that therapeutics in UTIs may be suboptimal, potentially leading to therapeutic failure and increased bacterial resistance.

### Methods

This study aimed to analyze the antibiotic prescription patterns in adult patients with suspected UTIs and to evaluate the appropriateness of the antibiotic prescription. This is a cross-sectional study of patients treated in outpatient centers and in a second-level hospital of the Ministry of Public Health (MOPH) in a city in Ecuador during 2019. The International Classification of Disease Tenth Revision (ICD-10) was used for the selection of the acute UTI cases. The patients included in this study were those treated by family, emergency, and internal medicine physicians.

### Results

We included a total of 507 patients in the analysis and 502 were prescribed antibiotics at first contact, constituting an immediate antibiotic prescription rate of 99.01%. Appropriate criteria for antibiotic prescription were met in 284 patients, representing an appropriate prescription rate of 56.02%. Less than 10% of patients with UTI had a urine culture. The most frequently prescribed antibiotics were alternative antibiotics (also known as second-line antibiotics), such as ciprofloxacin (50.39%) and cephalexin (23.55%). Factors

Infection in Ecuador", Mendeley Data, V1, doi: 10.
17632/styystf2nj.2 URL: https://data.mendeley.
com/datasets/styystf2nj/2.

**Funding:** The author(s) received no specific
funding for this work.

**Competing interests:** The authors have declared
that no competing interests exist.

associated with inappropriate antibiotic prescribing for UTIs were physician age over forty
years, OR: 2.87 (95% CI, 1.65–5.12) *p*<0.0001, medical care by a general practitioner,
OR: 1.89 (95% CI, 1.20–2.99) *p* = 0.006, not using point-of-care testing, OR: 1.96 (95%
CI, 1.23–3.15) *p* = 0.005, and care at the first level of health, OR: 15.72 (95% CI, 8.57–
30.88) *p*<0.0001.

## Conclusions

The results of our study indicate an appropriate prescription rate of 56.02%. Recommended
antibiotics such as nitrofurantoin and fosfomycin for UTIs are underutilized. The odds for
inappropriate antibiotic prescription were 15.72 times higher at the first level of care com-
pared to the second. Effective strategies are needed to improve the diagnosis and treatment
of UTIs.

## Introduction

Urinary tract infections (UTI) are the most prominent urological illness among men and
women, and they are one of the most common infections in primary care [1, 2]. It is estimated
that almost half of patients with UTIs go to primary care clinics and a quarter to emergency
departments [3]. UTIs may be caused by several organisms; *Escherichia coli* remains the pri-
mary agent responsible for UTIs in both outpatient and inpatient settings. [4–6]. Other uro-
pathogens involved are *Enterococcus faecalis*, *Enterobacter spp.*, *Staphylococcus saprophyticus*,
*Klebsiella pneumoniae*, *Proteus mirabilis*, and *Pseudomonas spp.* [7].

UTIs are clinically manifested through a variety of signs and symptoms such as dysuria, fre-
quency, and urgency of urination. About 25% of women who encounter a first episode of UTI
suffer a recurrence within 6 months [8]. Currently, the diagnosis relies on clinical symptoms
coupled with findings from laboratory examinations, including the detection of bacteria in
urine via nitrite strips and the semi-quantitative assessment of white blood cell count in urine
[9, 10]. Despite being the conventional method for diagnosing UTI, urine culture is costly and
time-intensive [11].

Most UTI cases are uncomplicated and in clinical practice do not require sophisticated lab-
oratory tests for diagnosis [10, 12]. However, suboptimal diagnostic procedures can lead to
over- or undertreatment that can in turn lend themselves to serious and potentially life-threat-
ening complications. Overtreatment may be the result of empirical antibiotic treatment based
on symptoms alone or incorrect interpretation or classification of current Point-Of-Care
(POC) diagnostic test results [13, 14]. This may be explained by differences in the decision-
making process of physicians in treating UTI.

UTIs are the leading cause for antibiotic prescriptions following respiratory tract infections
[15–18]. Uncomplicated cases can be treated empirically with antibiotics; however, clinical
practice variation in treating UTIs has been observed [19–21]. The disparity in the diagnosis
and prescription of drugs for different diseases could result in wasted resources, exposing
patients to an unnecessary risk of adverse events and, in the case of antibiotic prescriptions, to
a risk of antimicrobial resistance [22, 23]. According to several studies, the prevalence of multi-
drug-resistant uropathogens is increasing and current therapeutics used to treat UTIs may be
suboptimal [24–28]. This can be attributed in part to the unnecessary prescription of antibiot-
ics, but also to the use of antibiotics other than those recommended.

## Methods

Ethics approvals for the protocol and the study were granted by the Subcommittee for Research Ethics on Human Beings–Pontificia Universidad Católica del Ecuador with authorization code SB-CEISH-POS-694 dated March 25, 2021. We meticulously adhered to all relevant guidelines and national regulations throughout our methodology. Written informed consent was procured from every participant. This process encompassed presenting participants with an in-depth outline of the study's objectives, procedures, potential risks, and benefits. During this process, participants had the opportunity to inquire about any aspects of the study. Their voluntary agreement to utilize their clinical information from the Electronic Health Record (HER) for research purposes was documented in the informed consent forms. Prior to signing, participants received thorough explanations of the contents of the consent forms, and their signatures were witnessed by a trained staff member.

### Design and setting

Th study aimed to analyze the antibiotic prescription patterns in adult patients with suspected acute UTIs in Ecuador and to evaluate the appropriateness of antibiotic prescriptions in the first and second level of healthcare. In Ecuador, the first level of healthcare is represented by outpatient facilities, staffed by generalist doctors such as family physicians, who serve as the closest initial point of contact with the population. Meanwhile, the second level of healthcare is provided in medium-complexity hospital settings where both outpatient and inpatient services are offered. These services are attended to by both generalist doctors and specialists from other fields [29]. The design of this cross-sectional study is similar to other studies, allowing us to obtain comparable results [30–32].

### Data source

The data source for the variables included in this study was the EHR of patients from 14 outpatient centers (8 urban and 6 rural) of the first level of healthcare, and from one second-level hospital of the Ministry of Public Health (MOPH) in Ibarra, Ecuador during 2019. These centers have been using EHR since 2010. Physicians enter the information directly into the EHR on a computer during the outpatient appointment or on rounds during inpatient care. The information was manually extracted from the EHR between March 27, 2021 and June 27, 2021. The selection of acute UTI cases was based on the International Classification of Diseases Tenth Revision (ICD-10). Two peer reviewers autonomously extracted data, adhering to the specified inclusion criteria: UTI cases identified by ICD-10 codes N10: Acute pyelonephritis, N300: Acute cystitis, and N390: Urinary tract infection, site not specified; individuals aged 18 and above who had been diagnosed with acute UTI and received treatment within the MOPH system, whether as outpatients or inpatients; and patients under the care of family, emergency, and internal medicine physicians. Exclusions comprised pregnant patients, those managed by other healthcare subsystems, and chronic UTI diagnoses including N30.1: Interstitial cystitis (chronic), N30.2: Other chronic cystitis, N11.0: Nonobstructive reflux-associated chronic pyelonephritis, and N11.1: Chronic obstructive pyelonephritis.

Upon completion of individual data extraction, a thorough comparison was conducted, leading to consensus decisions regarding the inclusion or exclusion of each patient.

### Sample

To establish the sample, we utilized the total reported instances of UTI in the adult population by the MOPH for Ibarra in 2019. A probabilistic sampling method was employed to ensure

representative selection. The sample size was determined from the cumulative count of UTI cases officially documented by the MOPH in Ibarra for 2019, encompassing 3,475 cases from outpatient centers and 273 cases from the second-level hospital. To address variance in case dispersion among healthcare facilities, a stratified sampling technique was employed, considering the proportion of UTI cases treated at each facility in the sample selection process for both settings. Random selection was executed using a random number system to ensure impartiality. This method aimed to secure a sample that faithfully mirrored the distribution of UTI cases across diverse facilities and settings. The following formula was applied to calculate the sample for a finite universe: $n = N*Z^2*p*q / d^2*(N-1) + Z^2*p*q$, where $N$ is population size, $Z$ is the confidence level (95%), $p$ is the expected proportion (50%) due to an unknown proportion of inappropriate antibiotic prescription, $q$ is the probability of failure (50%), and $d$ is precision (5% of maximum admissible error in terms of proportion). The sample was defined as the patients who met the inclusion criteria and that had complete information in their EHR. Additionally, the sample was ensured not to include repeated measurements within the same patient. In the end, our sample was comprised of 507 patients, 347 coming from outpatient centers and 160 from the second level hospital.

## Assessment of appropriateness of antibiotic prescription

In this study, the appropriateness of antibiotic prescription was delineated as the imperative requirement for immediate antibiotic usage, aligning with the patient's characteristics and reported symptoms, as well as the chosen type of antibiotic. We assessed appropriateness based on the subsequent criteria: [12, 33–35]:

- any woman with at least two typical urinary symptoms (dysuria, frequency or urgency to urinate) and no genital symptoms (genital discharge).

- any patient with any typical symptoms of UTI and lumbar pain or fever.

- any patient with typical symptoms of UTI and comorbidities (immunosuppression or reported anatomic urinary tract disease).

- any male with typical symptoms of UTI with no urethral discharge.

- any patient with UTI symptoms and a positive urine test result for UTI.

The purpose of measuring the appropriate prescription rate was to identify the over-prescription that could be occurring. Additionally, these criteria facilitated the categorization of UTI diagnoses based on clinical parameters, allowing for the differentiation between acute cystitis and acute pyelonephritis.

## Statistical analysis

The variables included in this study were the characteristics of the patients and prescribers. We performed a descriptive analysis of qualitative variables through frequency distributions and proportions, and of quantitative variables through measures of central tendency and dispersion. The description and definitions of the variables included in the study can be found in supplementary material in S1 Table. A mixed effects logistic regression model was developed using the "lme4" package [36] in R (version 4.2.2) to analyze the appropriateness of antibiotic prescription. The dependent variable, "Inappropriateness of Antibiotic Prescription," was regressed against predictor variables. Candidate independent variables were selected based on theoretical relevance and refined through preliminary univariate logistic regression models. Collinearity among variables was assessed using variance inflation factors (VIF), revealing no

significant concerns. The final model was represented by the equation:

$$Inappropriate\_prescription = \beta_0 + \beta_1(Prescriber\_age) + \beta_2(Health\_professional) +$$
$$\beta_3(Lab\_requested) + \beta_4(Level) + u_{Health\_professional} + \varepsilon$$

Where *Inappropriate_prescription* represents the dependent variable. $\beta_0$ is the intercept of the model, $\beta_1, \beta_2, \beta_3, \beta_4$ are the coefficients for the predictor variables *Prescriber's age*, *Health professional*, *Lab test requested*, and *Level of healthcare*, respectively. $u_{Health\_professional}$ represents the random effect for *Health professional* (prescribers), and $\varepsilon$ is the error term. The binomial distribution and a logit link function were employed to model the probability of the dependent variable. Model selection was guided by the Akaike Information Criterion (AIC), balancing model complexity and fit to ensure the derivation of a well-optimized logistic regression model, effectively addressing overfitting concerns and enhancing the robustness of our findings.

## Results

The general characteristics of our sample are described in Table 1.

A total of 507 patients met the inclusion criteria for the study. Of the total number of UTI cases, 68.44% (347/507) were treated in the first level of health care and 31.56% (160/507) in the second level of health care. The patients with a UTI diagnosis were treated by 48 health professionals (37 general practitioners and 11 specialists).

Of the total number of UTI cases, 89.74% (455/507) were in females; the age groups with the most UTI cases in females were 18–29 years and over 50 years, 31.65% (114/455) and 32.31% (147/455), respectively. Only 10.26% (52/507) of UTI diagnosis were reported in males; 48.08% (25/52) were in the under-40 age groups, and 40.38% (21/52) were in males over 50 years of age.

Comorbidities were present in 30.37% (154/507) patients, with the most frequent being gastritis 38.96% (60/154) and hypertension 28.57% (44/154) (Table 1). The ICD-10 diagnostic codes registered were acute pyelonephritis (N10) in 11.64% (59/507) of patients, acute cystitis (N300) in 6.51% (33/507) of patients, and urinary tract infection of an unspecified site (N390) in 81.85% (415/507) of patients.

The symptoms more frequently reported were dysuria 80.07% (406/507), frequency 61.53% (312/507), urgency 50.09% (254/507), lumbar pain 23.66% (120/507), genital symptoms 5.92% (30/507), fever 3.35% (17/507), and macrohematuria 0.79% (4/507). Using clinical criteria for UTI diagnosis, 72.78% (369/507) of cases were categorized as acute cystitis, and 27.21% (138/507) identified as acute pyelonephritis.

Laboratory tests were requested in 74.16% (376/507) of patients, 62.24% (216/347) at the first level of care and in all patients at the second level of care. Of the total laboratory tests requested, urine dipsticks accounted for 98.93% (372/376). The results of the dipsticks included reactions for nitrites and leucocytes; reactions were positive for leucocytes in 92.47% (344/372) cases, for nitrites in 53.49% (199/372) and for both tests in 52.95% (197/372). Microscopy accounted for 7.44% (28/376) of the laboratory tests ordered, all of them after a previous urine dipstick test. Microscopy results included the identification of the quantity of bacteria per high-power field and Gram staining in all cases.

Urine culture accounted for 10.64% (40/376) of the laboratory tests ordered. Of the total number of urine cultures, 92.50% (37/40) were requested in the second level of health care (hospital). Based on clinical diagnosis, 57.5% (23/40) were ordered in acute pyelonephritis cases and 42.50% (17/40) in acute cystitis cases. Three urine cultures were requested at the first

**Table 1. Characteristics of the sample.**

| Prescribers | Total (%) |
|---|---|
| **Total prescribers** | 48 (100) |
| **Age (SD)** | |
| Mean | 37.6 (8.9) |
| **Gender** | |
| Female | 28 (58.33) |
| Male | 20 (41.67) |
| **Health Professionals** | |
| General Practitioners | 37 (77.08) |
| Specialists | 11 (22.92) |
| **Patients** | **Total (%)** |
| **Total patients** | 507 (100) |
| **Age** | |
| Mean (SD) | 41.9 (17.2) |
| **Age** | |
| 18–29 years | 157 (30.97) |
| 30–39 years | 111 (21.89) |
| 40–49 years | 71 (14.00) |
| ≥ 50 years | 168 (33.14) |
| **Gender** | |
| Female | 455 (89.74) |
| Male | 52 (10.26) |
| **Education** | |
| Primary | 107 (21.10) |
| High school | 339 (66.86) |
| University | 61 (12.03) |
| **Sexual activity in the past 12 weeks** | |
| Yes | 366 (72.19) |
| No | 141 (27.81) |
| **Comorbidities** | |
| Yes | 154 (30.37) |
| No | 353 (69.63) |
| **Type of comorbidities** | |
| Gastritis | 60 (38.96) |
| Hypertension | 44 (28.57) |
| Hypothyroidism | 11 (7.14) |
| Diabetes Mellitus | 7 (4.55) |
| Hypoacusis | 5 (3.24) |
| Pterygium | 5 (3.24) |
| Arrythmias | 4 (2.59) |
| History of Asthma | 4 (2.59) |
| History of Rheumatic Fever | 4 (2.59) |
| Parkinson | 3 (1.94) |
| Benign Prostatic Hyperplasia | 2 (1.29) |
| Urolithiasis | 2 (1.29) |
| Paraplegia | 2 (1.29) |
| Cystocele | 1 (0.65) |
| **Level of Healthcare** | |

(*Continued*)

**Table 1.** (Continued)

| | |
|---|---|
| First level of health care | 347 (68.44) |
| Second level of health care (hospital) | 160 (31.56) |
| **Type of consultations** | |
| General Practitioner consultations | 279 (55.03) |
| Specialist consultations | 228 (44.97) |

level of health care, two for the diagnosis of acute cystitis and one for the diagnosis of acute pyelonephritis.

The results of all urine cultures were positive for uropathogenic bacteria (>100,000 CFU) in 62.50% (25/40) of cases, of which 84% were *Escherichia coli* (33.3% were extended spectrum beta-lactamase—ESBL), 8% *Klebsiella pneumoniae*, and 8% *Proteus mirabilis*. Antibiogram results indicated that 52% (13/25) of isolated agents were resistant to ciprofloxacin, 56.52% to second generation oral cephalosporins (cefuroxime), 58.33% to Trimethoprim-sulfamethoxazole (TMP/SMX), 45.45% to third generation cephalosporins (ceftriaxone), 36.84% to ampicillin/sulbactam, 21.73% to nitrofurantoin, 14.28% to fosfomycin, 30.43% to gentamicin, and 0% to amikacin.

## Antibiotic prescription rate and appropriate antibiotic prescription rate

Among the 507 patients, 502 were prescribed antibiotics at first contact, constituting an immediate antibiotic prescription rate of 99.01%. Appropriate criteria for antibiotic prescription were met in 284 patients, representing an appropriate prescription rate of 56.02%.

Antibiotic use by type of clinical diagnosis is exhibited in Table 2. The mean duration of antibiotic treatment was 7 days (range 3 to 14 days). For all diagnoses, and at both levels of care, the most prescribed antibiotic was ciprofloxacin 50.40% (253/502), followed by cephalexin 23.51% (118/502), and nitrofurantoin 18.33% (92/502). In contrast, cefuroxime, fosfomycin, and aminoglycosides (gentamicin and amikacin) were the least prescribed antibiotics. Ceftriaxone, meropenem, imipenem, and piperacillin tazobactam were only prescribed at the second level of health care in cases of acute pyelonephritis.

In females, most prescribed antibiotics were fluoroquinoles 49.11% (221/450) and cephalosporins 26.67% (120/450), whereas in male cases fluoroquinoles dominated with 61.54% (32/52), followed by prescriptions of cephalosporins 21.15% (11/52) (Fig 1).

The results of the univariate regression analyses for inappropriate antibiotic prescription are documented in Table 3. Table 4 presents our definitive multiple logistic regression model, highlighting the influence of factors such as prescriber's age, type of health professional, and the laboratory test requested during first contact. This analysis reveals that the adjusted odds for inappropriate antibiotic prescription were notably higher for the first level of healthcare compared to the second level, OR: 15.72 (95%CI, 8.57–30.88) p<0.0001.

## Discussion

UTI is one of the most common bacterial infections in all levels of care and the incidence in women is much higher than in men. Our findings indicate that nearly 90% of patients seeking evaluation for UTI were women, with a higher prevalence observed among women aged 18–29 years. This pattern of distribution is consistent with similar studies in the field [37–40]. We also found that the group of women over 50 years of age were frequently diagnosed with UTIs,

**Table 2. Antibiotic use by type of clinical diagnosis.**

| Type of antibiotic | Female | | Male | | Total (%) |
|---|---|---|---|---|---|
| | Cystitis (%) | Pyelonephritis (%) | Cystitis (%) | Pyelonephritis (%) | |
| Ciprofloxacin | 165 (65.22) | 56 (22.12) | 26 (10.28) | 6 (2.37) | 253 (50.40) |
| Cephalexin | 80 (67.80) | 30 (25.42) | 6 (5.08) | 2 (1.69) | 118 (23.51) |
| Nitrofurantoin | 69 (75.00) | 20 (21.74) | 2 (2.17) | 1 (1.09) | 92 (18.33) |
| Ceftriaxone | 0 (0) | 9 (75.00) | 3 (25.00) | 0 (0.00) | 12 (2.39) |
| TMP/SMX | 2 (28.57) | 2 (28.57) | 3 (42.86) | 0 (0.00) | 7 (1.39) |
| Amoxicillin clavulanate | 3 (75.00) | 1 (25.00) | 0 (0.00) | 0 (0.00) | 4 (0.80) |
| Ampicillin sulbactam | 1 (25.00) | 2 (50.00) | 1 (25.00) | 0 (0.00) | 4 (0.80) |
| Meropenem | 0 (0.00) | 2 (66.67) | 1 (33.33) | 0 (0.00) | 3 (0.60) |
| Amoxicillin | 0 (0.00) | 0 (0.00) | 1 (50.00) | 0 (0.00) | 2 (0.40) |
| Imipenem | 0 (0.00) | 1 (100) | 0 (0.00) | 0 (0.00) | 2 (0.40) |
| Piperacillin tazobactam | 0 (0.00) | 1 (100) | 0 (0.00) | 0 (0.00) | 1 (0.20) |
| Cefuroxime | 0 (0.00) | 1 (100) | 0 (0.00) | 0 (0.00) | 1 (0.20) |
| Fosfomycin | 0 (0.00) | 1 (100) | 0 (0.00) | 0 (0.00) | 1 (0.20) |
| Gentamicin | 0 (0.00) | 1 (100) | 0 (0.00) | 0 (0.00) | 1 (0.20) |
| Amikacin | 0 (0.00) | 1 (100) | 0 (0.00) | 0 (0.00) | 1 (0.20) |

**TMP/SMX**: Trimethoprim-sulfamethoxazole

similar to the results reported by Wong, *et al.* [41] in which 33.6% of the total cases were found in the group of 51–64 years.

Symptomatic UTIs are rare in healthy young men; however, we found them in almost 10% of the cases. Sexual activity may be a risk factor for development of UTIs in healthy young men [42]; among the men in our study 71.15% had been sexually active within the previous twelve weeks. On the other hand, it has been shown that after middle age, the incidence of UTI increases progressively in men, and after 50 years of age the incidence is similar to that of women [40, 43, 44]. This considerable increase is mainly due to the consequences of prostate enlargement, common in older men; however, this antecedent was only reported in two cases, and it is possible that there was an underreporting of prostatic hyperplasia cases in our study.

In our study, general practitioners were responsible for a slightly greater number of UTI diagnoses compared to specialist physicians. This is consistent with a study conducted in Germany in outpatients during 2015–2019 [45], where UTI diagnoses and drug prescriptions were almost equal among these types of professionals. This could be explained by similarities between the health systems of the two countries; in Ecuador, the second level of health care may still include general practitioners. That same study also used ICD-10 diagnostic codes for the classification of UTIs and reported 64.3% of urinary tract infection of an unspecified site (N390), which is significant, but lower than that recorded in our study. In outpatient settings, not all technologies are always available to reach an accurate diagnosis. Moreover, there may be difficulty in achieving an accurate diagnosis only using the clinical features reported by patients, which may explain the way physicians code.

Clinical and laboratory features in patients with suspected UTIs can aid in the diagnosis. Dysuria, leukocytes above a trace amount, and positive nitrites have been shown to be strongly associated with a positive urine culture result and could be implemented as a strategy to reduce the use of unnecessary tests [46]. We found that dysuria was the most frequently reported symptom (80.07%), and that urine dipstick was the predominant test used for the diagnosis of UTIs (98.93%); comparable results have been published in other studies [19, 47, 48].

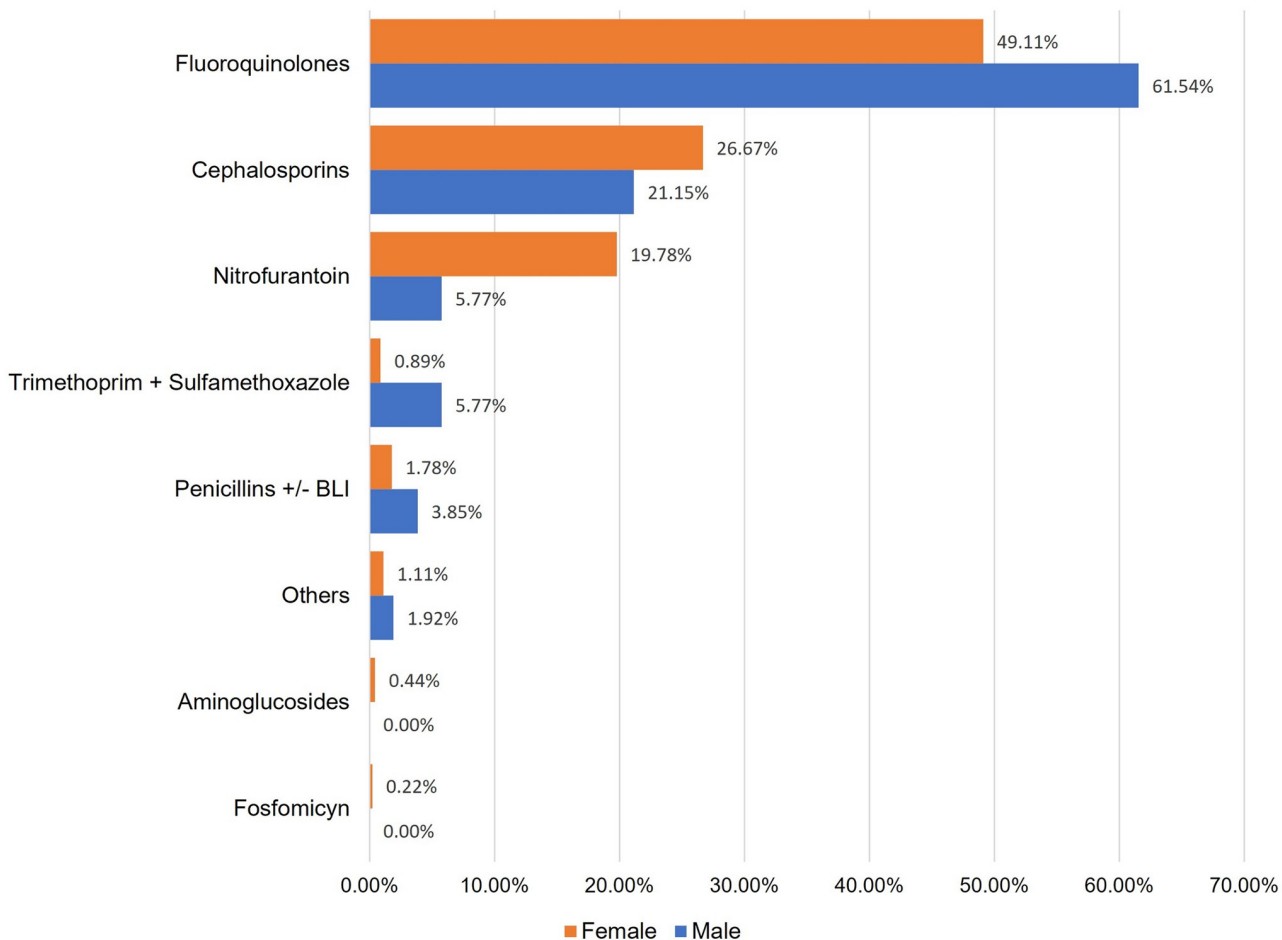

**Fig 1. Antibiotic use by sex. BLI**: Beta-lactamase inhibitors.

In the assessment of uncomplicated UTIs in outpatients, routine bacterial urine cultures might not always be essential, as they may not change the selected treatment strategy and may result in increased expenses [49]. An important classification of uncomplicated versus complicated UTIs distinguishes the need for urine culture [50]. We observed that less than 10% of patients with UTIs had a urine culture; lower results than those reported in Ireland, Spain, and Colombia in primary care, with 21%, 37% and 44% respectively [48, 51, 52]. One reason for these findings is the lack of availability of microbiological laboratory resources such as cultures in primary care in MOPH centers in Ecuador. This explains why almost all of these tests were performed at the second level of care. Moreover, we found a low percentage of positivity in urine cultures. This could be explained by an inadequate handling of the samples, an initiation of antibiotic therapy prior to taking the samples due to self-medication by the patients, or a deficient selection of the samples that require a microbiological study, since it has been demonstrated that a positive result for leukocytes in the test strip as a predictor of bacterial growth has a lower performance than the bacterial count [53, 54]. In addition, urine cultures were positive based on the commonly used threshold of >100,000 CFU/ml. It is important to acknowledge that this threshold is a widely accepted criterion for diagnosing UTI. However, it's worth

**Table 3. Associated factors for inappropriate antibiotic prescription.**

| Variable | Category | OR | 95%CI | p value |
|---|---|---|---|---|
| Level of healthcare | First level | 15.79 | 8.76–28.46 | <0.0001 |
| | Secondary (REF) | 1 | - | - |
| Prescriber's gender | Male | 0.85 | 0.59–1.21 | 0.849 |
| | Female (REF) | 1 | - | - |
| Prescriber's age | ≥ 40 years | 1.11 | 0.75–1.66 | 0.581 |
| | < 40 years (REF) | 1 | - | - |
| Health professional | General Practitioner | 1.27 | 0.89–1.80 | 0.190 |
| | Specialist (REF) | 1 | - | - |
| Patient's gender | Male | 0.78 | 0.43–1.39 | 0.398 |
| | Female (REF) | 1 | - | - |
| Patient's age | 18–29 years | 1.31 | 0.84–2.04 | 0.239 |
| | 30–39 years | 1.56 | 0.96–2.54 | 0.072 |
| | 40–49 years | 2.21 | 1.26–3.88 | 0.006 |
| | ≥ 50 years (REF) | 1 | - | - |
| Patient's educational level | Primary | 1.59 | 0.85–3.00 | 0.150 |
| | High school | 0.94 | 0.54–1.62 | 0.813 |
| | University (REF) | 1 | - | - |
| Sexual activity | Yes | 1.76 | 1.19–2.66 | 0.005 |
| | No (REF) | 1 | - | - |
| Comorbidity | Yes | 0.72 | 0.49–1.05 | 0.09 |
| | No (REF) | 1 | - | - |
| Lab test requested | No lab test request | 4.41 | 2.86–6.79 | <0.0001 |
| | Lab test request (REF) | 1 | - | - |

**REF**: Reference category

noting that lower CFU/ml counts can also be clinically significant, particularly in cases of uncomplicated cystitis in women [55, 56].

At the primary level, there was no evidence of the presence of information on the bacterial etiological profile of UTIs, the purpose of which is to guide the appropriate selection of empirical antibiotic treatment. This is even more important at the out-of-hospital level, considering the difficulty in accessing timely laboratory tests to support the presumptive diagnosis of UTIs. At the hospital level, a broader approach is required, suggesting the implementation of a

**Table 4. Adjusted odds ratios for inappropriate antibiotic prescription.**

| Variable | Category | AOR | 95%CI | p value |
|---|---|---|---|---|
| Prescriber's age | ≥ 40 years | 2.87 | 1.65–5.12 | <0.0001 |
| | < 40 years (REF) | 1 | - | - |
| Health professional | General practitioner | 1.89 | 1.20–2.99 | 0.006 |
| | Specialist (REF) | 1 | - | - |
| Lab test requested | No lab test request | 1.96 | 1.23–3.15 | 0.005 |
| | Lab test request (REF) | 1 | - | - |
| Level of healthcare | Primary care level | 15.72 | 8.57–30.88 | <0.0001 |
| | Secondary (REF) | 1 | - | - |

**REF**: Reference category, **AOR**: Adjusted Odds Ratio

rational antibiotic use program to improve the clinical response of each patient, protect the available therapeutic arsenal, and minimize the development of resistance, both at the individual and community level [57, 58].

## Patterns of antibiotic prescription

Antibiotic usage in UTI cases displays significant variation; according to existing literature, the most frequently utilized antibiotics include ciprofloxacin (17%–36.4%), TMP-SMX (14.2%–26.3%), nitrofurantoin (20.4%–35.9%), fosfomycin (0.2%–16.1%), and cephalosporins (4.8%–20.1%) [31, 45, 59, 60]. This variation in antibiotic use can be attributed to the presence and adherence to local and international guidelines, individual prescriber preferences, patient-specific factors, and the local availability of antibiotics [61]. Additionally, local susceptibility patterns play a crucial role in antibiotic selection, particularly for common Gram-negative pathogens, guiding empirical antibiotic therapy for UTIs.

Nitrofurantoin and fosfomycin continue to be dependable and recommended choices for empiric treatment of acute uncomplicated cystitis. Among adult outpatient patients with UTIs, resistance of *E. coli* to both of these agents remains low [62–65]. However, our results show that nitrofurantoin was prescribed in less than 20% of cases and fosfomycin only in one case. On the other hand, nitrofurantoin was prescribed in 21 cases of clinical pyelonephritis, a situation where it is not recommended [66]. According to guidelines, fluoroquinolones (norfloxacin, ciprofloxacin, and levofloxacin) are considered alternative options, also known as second-line antibiotics, for uncomplicated cystitis. They may also be an appropriate choice for therapy in patients with pyelonephritis not requiring hospitalization, particularly when the overall resistance rate is less than 10% [66]. Despite the limited number of patients in our study who had urine culture and antibiogram results, our findings reveal that over half of the isolated agents demonstrated resistance to ciprofloxacin. These data are consistent with those reported by the national antimicrobial resistance surveillance network of Ecuador, which reports 50% intermediate susceptibility and 35% resistance to *E. coli* to ciprofloxacin in community-acquired UTI in the female population aged 15 to 60 years [62]. Our data indicate that ciprofloxacin was the most common antimicrobial agent prescribed in all cases of UTIs, which is similar to other studies [31, 45, 59, 60]. The factors influencing the choice of antibiotics by practitioners were not evaluated and are beyond the scope of our research.

Empirical therapy with TMP/SMX is no longer recommended for either outpatients with uncomplicated cystitis or those with complicated UTIs, owing to the elevated resistance rates observed in numerous regions [10, 67]. The resistance of *E. coli* to TMP/SMX experienced a notable rise, going from 17.2% to 22.2%, among adult female outpatients in the United States during the 2003–2012 timeframe [64]; our findings show that TMP/SMX was used in less than 10% of the UTI cases.

## Appropriate antibiotic prescription rate

Our results show an appropriate antibiotic prescription rate of 56.02%. This result is comparable to studies such as that of Chardavoyne, *et al.* [68] where the appropriate prescription rate was 68% in cystitis and 46% in pyelonephritis; that of Vellinga, *et al.* [51] with 55% rate of appropriate prescription in UTI, and that of Sigler, *et al.* [69] with 64.1% rate in uncomplicated UTIs. It is noteworthy to mention that the methodologies used to assess prescribing were different and the variability may be influenced by the distinct diagnostic criteria, the use of diagnostic support technologies, local antibiotic resistance, and antibiotic availability.

The odds of inappropriate antibiotic prescription were higher at the primary care level when cases were treated by general practitioners, when laboratory tests were not requested,

and prescribers were older than 40. Prescribing decisions in the presence of suspected UTIs can be complex and multifactorial. We believe that diagnostic uncertainty resulting from insufficient diagnostic capacity in primary care settings may drive inappropriate antibiotic use with UTIs. In accordance with the typology of healthcare units in Ecuador, diagnostic support services such as laboratory facilities, for instance, urine analysis, are only available in a few outpatient centers [29]. Microbiology laboratory services for urine cultures are exclusively accessible for outpatient and hospitalized patients in hospitals. This significantly constrains the diagnostic capability of first-level ambulatory healthcare centers [70]. Nevertheless, the country's healthcare system mandates transferring patients to higher complexity levels for laboratory tests, potentially causing delays in decision-making. This circumstance could impact treatment decisions made in the absence of diagnostic support. The lack of diagnostic possibilities related to the level of care has previously been identified as a general barrier to an adequate diagnostic process and treatment selection by physicians [71, 72]; moreover, the empirical use of antibiotics is more likely among general practitioners when there is diagnostic uncertainty [73, 74].

## Strengths and limitations

The main strength of this study is that the information was obtained directly from the EHR of each patient, which allows us to be confident in the data recorded. In addition, the information came from 14 urban and rural outpatient centers and one second-level hospital, which permitted us to obtain data that is quite representative, considering the variability of the clinical practice of almost 50 prescribers distributed in these centers. Additionally, the sample included was relatively large, yielding statistically significant results.

However, a significant limitation of our study stems from its reliance on retrospective data sourced from the EHRs. Firstly, the application of ICD-10 codes is widespread across EHRs, serving administrative, billing, and medical purposes for documenting diagnoses during hospitalization or outpatient visits. Regrettably, these codes are not specifically designed for research, potentially introducing errors of omission and misclassification, which could lead to an underestimation of case identification [75]. For example, the code N390 presented challenges in discerning whether certain prescriptions could be deemed appropriate, as it lacked disaggregation, making it difficult to differentiate between complicated and uncomplicated UTIs. However, existing evidence underscores the utility of employing ICD-10 UTI diagnosis codes, like those used in our study, for the identification of UTI cases [76]. Furthermore, while the use of both diagnosis and symptom codes enhances the detection of UTI cases within primary care internal medicine and family medicine settings, it is worth noting that symptom codes are scarcely utilized in medical practice within the Ecuadorian context. Secondly, it is conceivable that physicians may occasionally record symptoms or laboratory data based on their individual judgment, resulting in potential gaps within the records. Lastly, an additional limitation lies in our failure to scrutinize the decision-making process behind prescription practices, a factor that could potentially lead to an overestimation of inappropriate prescriptions. Nevertheless, it is important to emphasize that all variables considered in our study were consistently documented in the EHR of each included patient and factored into the subsequent analysis.

## Conclusion

This study identified inappropriate antibiotic prescription for UTIs in primary care. Although empirical antibiotic therapy is primarily used for the treatment of UTIs, inappropriate and excessive use of antibiotics is of concern. We noted a lack of use of recommended antibiotics,

such as nitrofurantoin and fosfomycin, in uncomplicated UTI cases, along with an excessive use of alternative antibiotics like fluoroquinolones and cephalosporins. These findings support the need for strategies to improve the diagnosis and treatment of UTIs. The adequate use of point-of-care testing in outpatients, better antibiotic selection, and continuous surveillance of local bacterial resistance by physicians could improve the appropriate prescription of antibiotics.

## Supporting information

**S1 Table. Variables and definitions.**
(DOCX)

## Author Contributions

**Conceptualization:** Xavier Sánchez, Alicia Latacunga, Iván Cárdenas.

**Data curation:** Xavier Sánchez.

**Formal analysis:** Xavier Sánchez, Alicia Latacunga, Iván Cárdenas.

**Investigation:** Xavier Sánchez, Alicia Latacunga, Iván Cárdenas, Ruth Jimbo-Sotomayor, Santiago Escalante.

**Methodology:** Xavier Sánchez, Alicia Latacunga, Iván Cárdenas.

**Project administration:** Xavier Sánchez.

**Supervision:** Xavier Sánchez.

**Validation:** Xavier Sánchez, Alicia Latacunga, Iván Cárdenas, Ruth Jimbo-Sotomayor, Santiago Escalante.

**Writing – original draft:** Alicia Latacunga, Iván Cárdenas, Ruth Jimbo-Sotomayor, Santiago Escalante.

**Writing – review & editing:** Xavier Sánchez, Ruth Jimbo-Sotomayor, Santiago Escalante.

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
