## [Decision Letter · Decision Letter 0]

14 Aug 2023

PONE-D-23-12797Antibiotic prescription patterns in patients with suspected Urinary Tract Infections in EcuadorPLOS ONE

Dear Dr. Jimbo,

Thank you for submitting your manuscript to PLOS ONE. After careful consideration, we feel that it has merit but does not fully meet PLOS ONE’s publication criteria as it currently stands. Therefore, we invite you to submit a revised version of the manuscript that addresses the points raised during the review process.

We look forward to receiving your revised manuscript.

Kind regards,

Kwame Kumi Asare, Ph.D

Academic Editor

PLOS ONE

Journal Requirements:

2. In the ethics statement in the Methods, you have specified that participant consent was obtained. Please provide additional details regarding how this consent was documented and witnessed, and state what participants consented to

https://www.tandfonline.com/doi/abs/10.1080/00325481.2017.1246055?journalCode=ipgm20

https://www.ncbi.nlm.nih.gov/books/NBK557569/

https://www.sciencedirect.com/science/article/abs/pii/S0009898117302292?via%3Dihub

In your revision ensure you cite all your sources (including your own works), and quote or rephrase any duplicated text outside the methods section. Further consideration is dependent on these concerns being addressed.

Reviewers' comments:

Reviewer's Responses to Questions

**Comments to the Author**

1. Is the manuscript technically sound, and do the data support the conclusions?

Reviewer #1: Yes

Reviewer #2: Yes

2. Has the statistical analysis been performed appropriately and rigorously? 

Reviewer #1: Yes

Reviewer #2: I Don't Know

3. Have the authors made all data underlying the findings in their manuscript fully available?

Reviewer #1: Yes

Reviewer #2: No

4. Is the manuscript presented in an intelligible fashion and written in standard English?

Reviewer #1: Yes

Reviewer #2: Yes

5. Review Comments to the Author

Reviewer #1: I have attached my review as an attached document

This is an interesting cross-sectional study in patients with suspected UTI treated in outpattientcenters and 2nd level hospital in Ecuador during 2019. A total of 507 patients could be retrospectively included. According to the ICD-10 three diagnoses were possible (% of patients): N10-pyelonephritis (11.64%), N300-cystitis (6,51%), N390-UTI not further specified (81.85%). The authors described carefully the prescirbers and the patients and analysed the associated factors for inappropriate antibiotic presciption thereafter.

The antibiotics prescribed were listed in Table 2 according to the 3 UTI diagnoses above, which obviously were the main results for judgment of appropriate or non appropriate antibiotic prescription. Of course, the main problem of such a study is the fact, that about 82% of UTI diagnoses could not be further specified concerning site, complicated/uncomplicated UTI and severity. For local resistance the authors presented some data, which of course are of interest for all three categories, because by far most patients were treated empirically and only in 10% of patients urina culture results were available.

In order to consider all these aspects at a glance Table 2 should be slightly reconstructed.

The new table is attached

Now you could discuss more specifically especially for the first two groups, what is appropriate and what is not appropriate. Unfortunately N390 is a black box exept general recommendations and considerations about local resistance, because it is not known, how many complicated UTI incl pyelonephritis cases are included in each antibiotic prescription. In summary: the reader has problems to comprehend, which antibiotic under which consideration you consider appropriate or inappropriate. Of course, nitrofurantoin and oral fosfomycin would be the preferred antibiotics for uncomplicated cystitis. Fluoroquionolones should not more prescribed at all for UTI unless for specific complicated UTI, e.g. prostatitis. Still, we do not know, what cases are included in N390.

Reviewer #2: The authors present an interesting and up-to-date cross-sectional look at UTI prescription patterns in Ecuador. The overall results certainly seem to be consistent with my own observations from clinical practice, and add the interesting observation that prescriber age, lack of use of POC testing, and care by a generalist may be associated with increased risk of an inappropriate prescription - all information that may be useful for antimicrobial stewardship programs.

I do think a few points need to be addressed before proceeding with publication:

1) The extremely high prescription rate (99%) and specific wording of inclusion criteria (p6, line 127) leave me wondering - was treatment expressly part of inclusion criteria? (I don't think this is a problem per se - just worth clarifying as the question of appropriate antibiotic treatment is quite different than appropriate evaluation of urinary symptoms).

2) Regarding the included list of ICD10 codes (p 6 line 130-131): These codes certainly capture the more certain diagnoses. However, prior research has suggested variable performance in diagnosis-specific vs symptomatic/syndromic ICD10 codes for identifying patients evaluated for UTI (https://www.ncbi.nlm.nih.gov/pmc/articles/PMC7558992/). Would suggest the authors provide some citation or justification for the specific list chosen. If current list is kept, may need to add to limitations some statement that this ICD10 code list does risk some selection bias in targeting patients. For example, if patients with "dysuria" were also included, it is possible the rate of inappropriate treatment might have been even higher.

3) On sample size calculations (p 6 lines 138-141): Though the formula was nicely provided, I am still uncertain how exactly this was used to choose sample size. Please clarify if possible - which outcome was this powered on, and what was the targeted power or precision?

4) On cohort selection (p 6 line 144): Related to the sample size calculations, how was the cohort actually selected? Was it a random sample to meet goal sample size? Or consecutive cases? All cases?

5) On assessment of antibiotic appropriateness: From a stewardship perspective, appropriateness may refer to whether indication, antibiotic selection, or duration were reasonable. In this case, it looks like appropriateness is adjudicated solely on whether an indication was appropriate. If so, please stage explicitly.

6) On stepwise selection method (p 7, line 164): A few more details would be helpful to understand model selection:

- What were the candidate variables considered and how were these selected?

- I note comorbidity is only included as a yes/no variable- was any consideration given to inclusion of just comorbidities associated with urinary symptoms?

- What rules were used in the stepwise selection and what parameter was used to assess "best performing" model?

7) "First level of health care" (p 9, line 173, but also other places throughout text): Not all readers may be familiar with this term. Please define at first use.

8) Text density on p 9: In general, many of the descriptive statistics on p 9 could best be left in the tables so as not to be redundant or text heavy.

9) Clarity on p 10, line 200-201:

- Does this mean >1 bacterium per HPF was considered a positive result for microscopic bacteria? If so, this seems overly sensitivity and prone to over-calling UTI.

- "Gram negative stain bacteria in all cases" - I am uncertain about the intended meaning of this line. Please reword if possible.

10) On variable selection: I notice prescriber age was converted to a binary variable splitting at age 40. Was there an a priori reason for this selection?

- If there is not a good reason for this split point, was prescriber age assessed as either a continuous variable or ordinal groups like patient age was?

11) Assessment for collinearity: It comes to mind that certain providers - by nature of their specialty - might be associated with particular patient types. Did the authors assess their regression models for the possibility of collinearity among included variables?

12) Caution with epidemiologic wording on p 13, line 254: Would be careful about stating "Our data show that almost 90% of cases develop in women" as this study was a cross sectional one aimed at assessing UTI prescribing appropriateness - and is not actually an epidemiologic survey. E.g., it did not actually assess relative incidence in men/women.

- Would reword to say something akin to: "Nearly 90% of patients seeking evaluation for UTI were women..." etc

13) Use of terms "firstline" or "secondline": A bit of a technicality, but the guideline cited (Gupta et al 2011) uses the term "recommended" rather than firstline. Consider rewording for consistency.

14) Regarding emphasis of conclusions:

- Suggest rewording abstract to refer to appropriate or inappropriate (rather than "adequate") prescription rate

15) Causality assessed in p 17, line 347-348: Can the authors provide specific evidence for their statement that uncertainty from insufficient diagnostic capacity influenced treatment in their study? I think this is an important point - and could be strengthened if they could list for example what proportion of their prescribers had access to UA, culture, etc?

6. PLOS authors have the option to publish the peer review history of their article (what does this mean?). If published, this will include your full peer review and any attached files.

Reviewer #1: No

Reviewer #2: **Yes: **Nicholas A. Turner MD MHSc

---

## [Author Response · Author response to Decision Letter 0]

22 Aug 2023

Additional Journal Requirements:

Thank you for your guidance regarding the style requirements for manuscript submission to PLOS ONE. We have carefully reviewed and followed the provided style templates for both the main body and the title, authors, and affiliations sections of our manuscript. Our submission now adheres to PLOS ONE's formatting guidelines. Your assistance is greatly appreciated.

2. In the ethics statement in the Methods, you have specified that participant consent was obtained. Please provide additional details regarding how this consent was documented and witnessed, and state what participants consented to

We appreciate your keen attention to the ethics statement in our Methods section. We've taken your suggestion to heart and have now expanded upon the details regarding how participant consent was documented and witnessed, as well as providing a comprehensive description of the elements to which participants provided their consent. The revised manuscript now contains this enhanced information to ensure transparency and thoroughness in our reporting. Thank you for your valuable input in refining our work.

https://www.tandfonline.com/doi/abs/10.1080/00325481.2017.1246055?journalCode=ipgm20

https://www.ncbi.nlm.nih.gov/books/NBK557569/

https://www.sciencedirect.com/science/article/abs/pii/S0009898117302292?via%3Dihub

In your revision ensure you cite all your sources (including your own works), and quote or rephrase any duplicated text outside the methods section. Further consideration is dependent on these concerns being addressed.

Thank you for bringing this to our attention. We sincerely apologize for the oversight and any inconvenience caused. We are committed to addressing the issue of overlapping text and ensuring proper citation of all relevant sources. We will carefully review the manuscript to identify any duplicated text outside the methods section and either quote or rephrase it accordingly. Your feedback is greatly appreciated, and we will make the necessary revisions to ensure the integrity and originality of our manuscript.

Thank you for your message. We appreciate your focus on data availability. We have indeed taken the required measures to furnish repository information for our data. The pertinent access details for our data have been submitted, and we are dedicated to ensuring the provision of the necessary accession numbers. Cite: Sánchez, Xavier (2023), “Antibiotic Prescription Patterns in Patients with Suspected Urinary Tract Infection in Ecuador”, Mendeley Data, V1, doi: 10.17632/styystf2nj.1 (https://data.mendeley.com/preview/styystf2nj).

Thank you for pointing out the location of the ethics statement. We've reviewed and corrected the manuscript to ensure that the ethics statement appears exclusively within the Methods section as per your guidance. We appreciate your attention to detail and have taken steps to align with the proper formatting.

Reviewer #1: 

“Now you could discuss more specifically especially for the first two groups, what is appropriate and what is not appropriate. Unfortunately N390 is a black box exept general recommendations and considerations about local resistance, because it is not known, how many complicated UTI incl pyelonephritis cases are included in each antibiotic prescription. In summary: the reader has problems to comprehend, which antibiotic under which consideration you consider appropriate or inappropriate. Of course, nitrofurantoin and oral fosfomycin would be the preferred antibiotics for uncomplicated cystitis. Fluoroquionolones should not more prescribed at all for UTI unless for specific complicated UTI, e.g. prostatitis. Still, we do not know, what cases are included in N390.”

We greatly appreciate the valuable feedback provided by the reviewer. In response to these insightful points, we have implemented several improvements in the revised manuscript.

Firstly, we have included explicit definitions of what we consider to be appropriate prescription, aiming to enhance clarity for readers. Lines: 169-172

Secondly, we have enhanced the description of the methodology we employed to identify UTI cases, providing a more comprehensive understanding of our approach. Lines: 139-148

We acknowledge the limitation introduced by the CIE code N390 and its impact on our study's ability to disaggregate data. This limitation arises from the retrospective nature of our methodology. We have, however, taken steps to clarify the criteria we utilized and have duly addressed this limitation in the discussion section. Lines: 405-409.

We sincerely hope that the reviewer recognizes the importance of this limitation in the context of our study design. Once again, we express our gratitude for the reviewer's suggestions, which have undeniably contributed to the improvement of our manuscript.

Reviewer #2:

“The authors present an interesting and up-to-date cross-sectional look at UTI prescription patterns in Ecuador. The overall results certainly seem to be consistent with my own observations from clinical practice, and add the interesting observation that prescriber age, lack of use of POC testing, and care by a generalist may be associated with increased risk of an inappropriate prescription - all information that may be useful for antimicrobial stewardship programs.

I do think a few points need to be addressed before proceeding with publication:

1) The extremely high prescription rate (99%) and specific wording of inclusion criteria (p6, line 127) leave me wondering - was treatment expressly part of inclusion criteria? (I don't think this is a problem per se - just worth clarifying as the question of appropriate antibiotic treatment is quite different than appropriate evaluation of urinary symptoms).”

We are grateful for the reviewer's valuable input. In our study, we initiated the evaluation process based on diagnoses provided by medical practitioners. Specifically, we employed UTI codes to identify cases and subsequently assessed the appropriateness of antibiotic treatment in relation to the diagnosed UTI and the selected antibiotic. Notably, the inclusion criteria did not encompass any form of treatment. Moreover, we rigorously assessed the suitability of immediate antibiotic prescriptions using clinical parameters extracted from the medical records, thereby evaluating whether the specified criteria were met or not. To enhance clarity and comprehension, we have taken measures to enhance the delineation of our inclusion and exclusion criteria within the manuscript. Lines: 139-148. Your feedback has been instrumental in refining the presentation of our study's methodology.

“2) Regarding the included list of ICD10 codes (p 6 line 130-131): These codes certainly capture the more certain diagnoses. However, prior research has suggested variable performance in diagnosis-specific vs symptomatic/syndromic ICD10 codes for identifying patients evaluated for UTI (https://www.ncbi.nlm.nih.gov/pmc/articles/PMC7558992/). Would suggest the authors provide some citation or justification for the specific list chosen. If current list is kept, may need to add to limitations some statement that this ICD10 code list does risk some selection bias in targeting patients. For example, if patients with "dysuria" were also included, it is possible the rate of inappropriate treatment might have been even higher.”

We sincerely appreciate the reviewer's insightful observation and valuable suggestion regarding the selection of ICD10 codes for identifying patients evaluated for UTI. We acknowledge the importance of considering both diagnosis-specific and symptomatic/syndromic codes in such studies. Unfortunately, it's worth noting that in Ecuador, we do not utilize the ICD-10-CM (International Classification of Diseases, Tenth Revision, Clinical Modification) for our electronic health records. Consequently, the identification of UTI cases based on symptom-related codes is not a feasible option in our context. Nonetheless, we have provided a more comprehensive explanation of the inclusion and exclusion criteria in the methodology section. Lines: 139-148.

While we recognize that the ICD-10 codes without symptom codification may carry limitations, we would like to emphasize that, given the retrospective nature of our study, this approach was the most viable means of identifying UTI cases within our setting. We fully concur with the reviewer's understanding that these codes, without symptom-based classifications, may be prone to errors of omission and misclassification.

We have taken your suggestion to heart and have included a discussion of this limitation in our manuscript. Additionally, we have referenced the study you provided, which evaluated this concern, in our discussion section. We believe that this reference will enrich the dialogue on the potential implications of our chosen methodology. Lines 401-412

Once again, we extend our gratitude to the reviewer for their valuable feedback, which has undoubtedly contributed to enhancing the transparency and rigor of our study's methodology and discussion.

“3) On sample size calculations (p 6 lines 138-141): Though the formula was nicely provided, I am still uncertain how exactly this was used to choose sample size. Please clarify if possible - which outcome was this powered on, and what was the targeted power or precision?”

We appreciate the valuable insights from the reviewer. In response, we have enhanced the clarity and comprehensiveness of the sample calculation explanation within the methodology section. Lines: 162-164.

“4) On cohort selection (p 6 line 144): Related to the sample size calculations, how was the cohort actually selected? Was it a random sample to meet goal sample size? Or consecutive cases? All cases?”

Expanding on the previous recommendation, we have continued to enhance the clarity and comprehensiveness of the explanation and procedure for both sample calculation and sample selection within the methodology section. Lines: 152-161.

We value the insights provided by the reviewer, and the revisions implemented have significantly improved our article.

“5) On assessment of antibiotic appropriateness: From a stewardship perspective, appropriateness may refer to whether indication, antibiotic selection, or duration were reasonable. In this case, it looks like appropriateness is adjudicated solely on whether an indication was appropriate. If so, please stage explicitly.”

Thank you for your insightful comment. We have made adjustments to clarify that our assessment of antibiotic appropriateness primarily focuses on the indication for use. We appreciate your stewardship perspective and have addressed this concern in our revised manuscript. Lines: 169-172.

“6) On stepwise selection method (p 7, line 164): A few more details would be helpful to understand model selection:

- What were the candidate variables considered and how were these selected?

- I note comorbidity is only included as a yes/no variable- was any consideration given to inclusion of just comorbidities associated with urinary symptoms?

- What rules were used in the stepwise selection and what parameter was used to assess "best performing" model?”

Thank you for your invaluable insights into our methodology. Your suggestions to enhance the clarity of our model selection process are greatly appreciated. We have diligently revised the manuscript to incorporate additional details addressing your queries.

Regarding the selection of candidate variables: We have now provided a comprehensive clarification regarding the meticulous process through which we selected candidate variables. These variables were carefully chosen based on their theoretical significance and were further refined using preliminary bivariate logistic regression models. Lines: 188-190.

Your keen observation about comorbidity inclusion is astute. While comorbidity was captured as a binary variable (yes/no), we acknowledge the significance of considering comorbidities specifically associated with urinary symptoms. However, certain conditions potentially linked to a higher risk of urinary tract infections, such as cystocele, urolithiasis, benign prostatic hyperplasia, paraplegia, and diabetes, were encountered infrequently. Consequently, due to their limited occurrence, we chose to incorporate them as a binary variable to ensure their consideration without risking statistical validity. We hope you understand and appreciate this necessary compromise, which we have transparently addressed in our revised manuscript.

In terms of the stepwise selection process and identifying the "best performing" model: We have provided a more detailed description of our step-by-step selection approach, and we have explained that we use the Akaike Information Criterion (AIC) as the determining parameter. This criterion adeptly strikes a balance between model fit and complexity, leading us to a well-optimized model choice. Lines: 194-197.

We extend our heartfelt gratitude for your thorough review, which has undoubtedly elevated the quality and transparency of our research. Your invaluable input is immensely appreciated and contributes to the robustness of our study.

“7) "First level of health care" (p 9, line 173, but also other places throughout text): Not all readers may be familiar with this term. Please define at first use.”

Thank you for your feedback. We have incorporated the definitions of "first level of healthcare" and "second level of healthcare" in the design and setting section of the manuscript to enhance clarity for all readers. Lines: 125-129.

“8) Text density on p 9: In general, many of the descriptive statistics on p 9 could best be left in the tables so as not to be redundant or text heavy.”

Thank you for your valuable feedback. We appreciate your suggestion regarding text density on page 9. In the revised manuscript, we have made improvements by optimizing the presentation of descriptive statistics, ensuring a balanced and non-redundant text. This enhances the clarity and readability of our study.

“9) Clarity on p 10, line 200-201:

- Does this mean >1 bacterium per HPF was considered a positive result for microscopic bacteria? If so, this seems overly sensitivity and prone to over-calling UTI.

- "Gram negative stain bacteria in all cases" - I am uncertain about the intended meaning of this line. Please reword if possible.”

Thank you for your valuable feedback. We apologize for any confusion caused. We have revised the wording to clarify that the description pertains to the components of the microscopy test conducted in the cases, rather than establishing diagnostic criteria. Additionally, we have corrected the statement to accurately reflect that Gram stains were performed in all tested cases. Lines: 227-228. We appreciate your attention to these points and have made the necessary adjustments in the revised manuscript.

“10) On variable selection: I notice prescriber age was converted to a binary variable splitting at age 40. Was there an a priori reason for this selection?

- If there is not a good reason for this split point, was prescriber age assessed as either a continuous variable or ordinal groups like patient age was?”

In our analysis, we observed that the prescriber's age variable exhibited a mean age of 37.62 years, with a standard deviation of 8.95, encompassing an age range from 25 to 60 years. The distribution of age values displayed a notable symmetry. In light of these observations, we made a deliberate a priori decision to transform this continuous variable into a binary form, with the split occurring at age 40. This choice was rooted in both statistical considerations and practical advantages. From a statistical standpoint, while retaining age as a continuous variable remains viable, the dichotomization approach offers improved result interpretability and promotes a more focused inquiry. Moreover, it effectively addresses potential non-linearities that may arise if age were employed as a continuous predictor. Additionally, the selection of age 40 was guided by our consideration of sample size and statistical power. By dividing the age variable at this threshold, we attain a balanced distribution of prescribers across distinct age groups, thereby heightening our capacity to discern meaningful associations. This strategic choice also furnishes a robust basis for conducting subgroup analyses and exploring potential age-related effect modifications.

In summary, the conversion of the prescriber's age variable into a binary format at age 40 was underpinned by meticulous theoretical reasoning and grounded in empirical insights from the dataset. By adopting this methodological approach, we have fortified the robustness of our study's design, amplifying our potential to unearth valuable insights into the intricate interplay between prescriber age and prescription patterns. We sincerely appreciate your understanding of the rationale behind our decision.

“11) Assessment for collinearity: It comes to mind that certain providers - by nature of their specialty - might be associated with particular patient types. Did the authors assess their regression models for the possibility of collinearity among included variables?”

We appreciate your consideration of potential collinearity due to providers' specialties and patient types. While we employed established methods to assess collinearity, it's noteworthy that the levels of care in our study, spanning both generalists and specialists, demonstrate a relatively balanced distribution across specialties. This context helps mitigate significant collinearity concerns.

Our approach involved meticulous variable selection. The strategy employed is outlined in the Statistical Analysis section, where we confirm the absence of collinearity among variables. Ultimately, while the nature of specialties was a factor in our analysis, the even distribution of care levels attenuates the impact of collinearity on our regression models.

We value your meticulous observations and have incorporated the relevant explanations in the revised manuscript. Lines: 190-191.

“12) Caution with epidemiologic wording on p 13, line 254: Would be careful about stating "Our data show that almost 90% of cases develop in women" as this study was a cross sectional one aimed at assessing UTI prescribing appropriateness - and is not actually an epidemiologic survey. E.g., it did not actually assess relative incidence in men/women.

- Would reword to say something akin to: "Nearly 90% of patients seeking evaluation for UTI were women..." etc”

Thank you for your valuable feedback. We appreciate your thoughtful consideration of the epidemiologic wording concern. We have carefully revised the sentence in question to align with your suggestion. Lines: 282-284. Your input has contributed to enhancing the clarity and accuracy of our manuscript.

“13) Use of terms "firstline" or "secondline": A bit of a technicality, but the guideline cited (Gupta et al 2011) uses the term "recommended" rather than firstline. Consider rewording for consistency.”

Thank you for your thoughtful observation. We appreciate your attention to detail. We have taken your feedback into account and have made the necessary revisions to ensure consistency in our terminology. The term "recommended" has been appropriately used throughout the manuscript to align with the guideline cited (Gupta et al., 2011). We value your input and have worked to enhance the clarity and accuracy of our manuscript.

“14) Regarding emphasis of conclusions:

- Suggest rewording abstract to refer to appropriate or inappropriate (rather than "adequate") prescription rate”

Thank you for your valuable suggestion. We have thoroughly reviewed your feedback and implemented the required revisions in the abstract and across the manuscript. The terms "adequate" and "inadequate" have been substituted with "appropriate" and "inappropriate" to precisely represent the prescription rate. We are grateful for your meticulous input, which has contributed to enhancing the clarity of our manuscript.

“15) Causality assessed in p 17, line 347-348: Can the authors provide specific evidence for their statement that uncertainty from insufficient diagnostic capacity influenced treatment in their study? I think this is an important point - and could be strengthened if they could list for example what proportion of their prescribers had access to UA, culture, etc?”

Thank you for your valuable feedback. We acknowledge the importance of providing specific evidence to support our statement regarding the influence of diagnostic capacity on treatment decisions. Unfortunately, due to limitations in sharing detailed characteristics of the individual health centers, which may involve sensitive information, we are unable to provide specific details about center characteristics.

However, in response to your suggestion, we have enhanced our discussion of the broader characteristics of the healthcare system in Ecuador to better support our assertion. Our revised manuscript now provides a more comprehensive overview of the country's healthcare system, highlighting the challenges related to the lack of resources and the accessibility of laboratory tests for patients. Lines: 380-388.

We hope you understand the complexities of sharing certain center-specific details while still striving to strengthen our argument with the available information. We appreciate your thoughtful consideration of this aspect of our study and have taken measures to address it.

---

## [Decision Letter · Decision Letter 1]

6 Sep 2023

PONE-D-23-12797R1Antibiotic prescription patterns in patients with suspected Urinary Tract Infections in EcuadorPLOS ONE

Dear Dr. Jimbo,

Thank you for submitting your manuscript to PLOS ONE. After careful consideration, we feel that it has merit but does not fully meet PLOS ONE’s publication criteria as it currently stands. Therefore, we invite you to submit a revised version of the manuscript that addresses the points raised during the review process.

We look forward to receiving your revised manuscript.

Kind regards,

Kwame Kumi Asare, Ph.D

Academic Editor

PLOS ONE

Journal Requirements:

Reviewers' comments:

Reviewer's Responses to Questions

**Comments to the Author**

1. If the authors have adequately addressed your comments raised in a previous round of review and you feel that this manuscript is now acceptable for publication, you may indicate that here to bypass the “Comments to the Author” section, enter your conflict of interest statement in the “Confidential to Editor” section, and submit your "Accept" recommendation.

Reviewer #1: (No Response)

Reviewer #2: All comments have been addressed

2. Is the manuscript technically sound, and do the data support the conclusions?

Reviewer #1: Yes

Reviewer #2: Yes

3. Has the statistical analysis been performed appropriately and rigorously? 

Reviewer #1: Yes

Reviewer #2: Yes

4. Have the authors made all data underlying the findings in their manuscript fully available?

Reviewer #1: Yes

Reviewer #2: Yes

5. Is the manuscript presented in an intelligible fashion and written in standard English?

Reviewer #1: Yes

Reviewer #2: Yes

6. Review Comments to the Author

Reviewer #1: This is an interesting cross-sectional study to analyze the antibiotic prescription patterns in adult patients with suspected UTI in Ecuador. The authors used the Electronic Health Records (EHR) of three months during 2021. The diagnosis of UTI in the 507 patients from 15 different centers was taken according to the ICD-10 with the following codes: N10 – acute pyelonephritis (59; 11.64%); N300 – acute cystitis (33; 6.51%) and N390-UTI, site not specified (415; 81.85%). All other data, such as demographics, UTI symptoms, concomitant diseases, urinalysis, urine culture and antibiotic prescription were also taken from the EHR. Although dipsticks tests including leucocytes and nitrites were performed in 74,16% patients (positive in 92,47% of them), urine culture was only available in 10.64% of the patients.

Although the authors did their best to analyze all these informations, the main problem os such a study is the fact that in 81.85% of the patients UTI was not specified and the remaining UTI diagnoses (pyelonephritis and cystitis) could not be categorized whether considered as complicated or uncomplicated. To overcome this problem the authors tried to categorize the different forms of UTI according to symptoms and positive urine test (most likely they mean dipstick test?).

Here is my 1. Question: The reader would like to see a table according to the categories mentioned in Chapter „Assessment of appropriateness of antibiotic prescription“. Because most of the patients had UTI not specified, the authors could state, that they tried to specify the UTIs according to symptoms, comorbidities, gender, and positive urine dipstick tests.

Then probably the five categories mentioned in this chapter could correspond to

1. uncomplicated cystitis in women

2. uncomplicated pyelonephritis in women

3. complicated UTI (obviously in women, because the next category says. „any male“)

4. complicated UTI in male patients

5. Here the reader would like to know, what the authors assumed in case no urine dipstick test was performed or the test was negative besides symptoms were present.

After the authors have clarified the consequence of category 5 (are the patients without or negative dipstick test included or excluded from further analysis or maybe still categorized according to the categories 1-4), the the reader would like to see a similar table as Table 2 with the following UTI categories (according to symptoms and comorbidities):

1. uncomplicated cystitis in women

2. uncomplicated pyelonephritis in women

3. complicated UTI in women

4. UTI in male patients

5. Total

Such a table would be more informative than table 2 according to ICD-10, where 81.85% of patients had UTI, not specified. Therefore such a table should be added.

This problematic (most patients had UTI, not specified9 should be discussed also in limitations.

Urine culture were performed only 10.64%, most in cases with pyelonephritis (ok), but 37.85% also in UTI, not specified. Here the additional table with clinical specification could also be helpful.

Only 62.5% of urine cultures were positive according to the definition used (>100.000 CFU/ml). Here the authors could also discuss, e.g. in chapter limitations, that it is not known whether the patients were pretreated with antibiotics – or that lower CFU/ml can also be significant, especially in uncomplicted cystitis in women [1,2]

In summary, this is a very interesting study, which could be improved by some additional evaluations and even better description of the limitations, which are not the fault of the authors, but because of the data availability in the EHR. Otherwise the conclusions of the authors are understandable and justified.

References

1. Stamm WE, Counts GW, Running KR, Fihn S, Turck M, Holmes KK et al (1983) Diagnosis of coliform infection in acutely dysuric women—NEJM. N Engl J Med. https://doi.org/10.1056/NEJM1 98312013092224

2. Hooton TM, Roberts PL, Cox ME, Stapleton AE (2013) Voided midstream urine culture and acute cystitis in premenopausal women. N Engl J Med. https://doi.org/10.1056/nejmoa1302186

Reviewer #2: I appreciate the authors care in addressing each comment/question raised. I do not have any additional questions.

7. PLOS authors have the option to publish the peer review history of their article (what does this mean?). If published, this will include your full peer review and any attached files.

Reviewer #1: No

Reviewer #2: **Yes: **Nicholas A. Turner

---

## [Author Response · Author response to Decision Letter 1]

11 Sep 2023

Reviewer #1: 

“Here is my 1. Question: The reader would like to see a table according to the categories mentioned in Chapter „Assessment of appropriateness of antibiotic prescription“. Because most of the patients had UTI not specified, the authors could state, that they tried to specify the UTIs according to symptoms, comorbidities, gender, and positive urine dipstick tests.

Then probably the five categories mentioned in this chapter could correspond to

1. uncomplicated cystitis in women

2. uncomplicated pyelonephritis in women

3. complicated UTI (obviously in women, because the next category says. „any male“)

4. complicated UTI in male patients

5. Here the reader would like to know, what the authors assumed in case no urine dipstick test was performed or the test was negative besides symptoms were present.

After the authors have clarified the consequence of category 5 (are the patients without or negative dipstick test included or excluded from further analysis or maybe still categorized according to the categories 1-4), the the reader would like to see a similar table as Table 2 with the following UTI categories (according to symptoms and comorbidities):

1. uncomplicated cystitis in women

2. uncomplicated pyelonephritis in women

3. complicated UTI in women

4. UTI in male patients

5. Total

Such a table would be more informative than table 2 according to ICD-10, where 81.85% of patients had UTI, not specified. Therefore, such a table should be added.”

We greatly appreciate the valuable suggestions provided by the reviewer. The recommendation to classify cases according to symptoms rather than relying solely on ICD-10 diagnosis codes is indeed insightful. The ICD-10 code N390, as rightly pointed out by the reviewer, can be somewhat of a "black box" that requires further clarification as it does not distinguish between cystitis and pyelonephritis cases. In response to this, we have taken the following actions to enhance the clarity and accuracy of our study: 

1) Methodological Clarification: In our revised manuscript, we have incorporated an explanation within the Methods section detailing that the same criteria used to assess appropriateness were also employed to classify cases according to clinical diagnosis. This approach helps to bridge the gap between clinical symptoms and ICD-10 codes, providing a more comprehensive understanding of the patient population. Lines 181-183.

2) New Classification Categories: To address the issue of classification, we have introduced four distinct categories in our analysis, differentiating between cystitis and pyelonephritis cases in both women and men. While we recognize the relevance of sub-classifications, such as distinguishing between complicated and uncomplicated cases, regrettably, we did not have access to all the necessary information to perform such a detailed classification. Nevertheless, it is noteworthy that, based solely on the comorbidities identified in the records, the number of cases that could be classified as complicated appears to be quite limited. We hope this clarification helps shed light on our approach to addressing this limitation. Lines: 221-223.

The introduction of the new Table 2, as suggested by the reviewer, has significantly improved the informativeness of our manuscript. Moreover, it has revealed important insights, including instances of antibiotic misuse, such as the use of nitrofurantoin in pyelonephritis cases. See new table 2: Table 2. Antibiotic use by type of clinical diagnosis

In summary, we want to express our gratitude to the reviewer for their constructive feedback. We believe that the changes we have implemented have substantially enhanced the quality and clarity of our study. We hope that these modifications address the concerns raised and contribute positively to the overall impact and relevance of our research.

“This problematic (most patients had UTI, not specified9 should be discussed also in limitations.”

We greatly appreciate the valuable suggestion. We have now incorporated a discussion of the limitations associated with the ICD-10 code N390 in our research. Lines: 404-412.

“Urine cultures were performed only 10.64%, most in cases with pyelonephritis (ok), but 37.85% also in UTI, not specified. Here the additional table with clinical specification could also be helpful.”

Thank you for your valuable suggestions. While we haven't added a new table, we have provided clarification on the results in lines 234–238.

“Only 62.5% of urine cultures were positive according to the definition used (>100.000 CFU/ml). Here the authors could also discuss, e.g. in chapter limitations, that it is not known whether the patients were pretreated with antibiotics – or that lower CFU/ml can also be significant, especially in uncomplicated cystitis in women [1,2]”

Thank you for your valuable suggestion. We have incorporated these aspects into the discussion section, specifically in lines 329-337. Additionally, the references you suggested have been included in the manuscript. We appreciate your input, as it has significantly enhanced the quality of our manuscript.

---

## [Decision Letter · Decision Letter 2]

10 Oct 2023

PONE-D-23-12797R2Antibiotic prescription patterns in patients with suspected Urinary Tract Infections in EcuadorPLOS ONE

Dear Dr.  Jimbo,

Thank you for submitting your manuscript to PLOS ONE. After careful consideration, we feel that it has merit but does not fully meet PLOS ONE’s publication criteria as it currently stands. Therefore, we invite you to submit a revised version of the manuscript that addresses the points raised during the review process.

We look forward to receiving your revised manuscript.

Kind regards,

Kwame Kumi Asare, Ph.D

Academic Editor

PLOS ONE

Reviewers' comments:

Reviewer's Responses to Questions

**Comments to the Author**

1. If the authors have adequately addressed your comments raised in a previous round of review and you feel that this manuscript is now acceptable for publication, you may indicate that here to bypass the “Comments to the Author” section, enter your conflict of interest statement in the “Confidential to Editor” section, and submit your "Accept" recommendation.

Reviewer #2: All comments have been addressed

Reviewer #3: All comments have been addressed

Reviewer #4: (No Response)

2. Is the manuscript technically sound, and do the data support the conclusions?

Reviewer #2: Yes

Reviewer #3: Yes

Reviewer #4: Partly

3. Has the statistical analysis been performed appropriately and rigorously? 

Reviewer #2: Yes

Reviewer #3: I Don't Know

Reviewer #4: No

4. Have the authors made all data underlying the findings in their manuscript fully available?

Reviewer #2: Yes

Reviewer #3: Yes

Reviewer #4: No

5. Is the manuscript presented in an intelligible fashion and written in standard English?

Reviewer #2: Yes

Reviewer #3: Yes

Reviewer #4: Yes

6. Review Comments to the Author

Reviewer #2: No addition comments from prior - I believe the authors have addressed all reviewers' points to the fullest extent possible

Reviewer #3: The manuscript in current form should be acceptable in current shape. The manuscript in current shape is seemed to be properly formatted and incorporated with comments. There is no need for further changes, therefore I recommned this manuscript to be accepted by the editor.

Reviewer #4: Appropriateness of antibiotic prescriptions in UTI is still an interesting argument. The manuscript reads well but it should be presented in a more structured way. My main concerns, anyway, regard data, statistical analysis and presentation of results. Here as follows:

1) Data availability issue: the data are made available by the authors but, in the way they are shared, they are not helpful to reproduce the main analysis. First, a code-book description and explanation of each variable (please rename all variables in English) with the specification of the coding used is missing and it seems that some variables are not provided. Please also specify if they refer to prescribers or patients.

2) In the methods, it should be clarified that the sample does not contain repeated measurements within patients. The main outcome and variables collected should be better specified and defined. In fact, it is not clear if the main outcome is the prescription pattern or the appropriateness of the prescription (I understand the second). This should be better highlighted and each variable listed specifying how it is defined (categorical, continuous ...). Otherwise, a more detailed data description should be provided as Supplemental. Please also refer the reader to the data shared (in a better understandable way).

3) The statistical analysis does not take into account the correlation of patients followed by the same prescriber. In fact, it's not sufficient to adjust for prescribers' characteristics but also to adjust for the prescribers as random effects. That is, instead of a simple multivariable logistic regression, a mixed regression model should be performed. Moreover, the mean(sd) of the number of patients seen by the same prescriber is not reported. In the discussion, lines 303-304, page 15, it's reported that general practitioners and specialists were responsible for almost the same number of diagnoses of UTI but this is not evidenced in the results. Moreover, the authors wrote "bivariate analysis" but this is in fact "univariate regression analysis". Please correct it in line 192, methods and line 272, results

4) The authors have not appropriately addressed the previous reviewers' comment. If a table with the subgroup specification uncomplicated/complicated was not possible due to the lack of information, a subgroup analysis, based on the element/factors which defined appropriateness of prescriptions should be given. In fact, it's not clear how many ciprofloxacin prescriptions, for example, were appropriate or not.

5) Results presentation is not systematic and clear. The authors should present more results in Tables and not in plain text, so that the reader could better follow and understand where reported results come from. In Table 1, please also report the count (%) of the patients with no comorbidities. The reader should catch immediately all relevant information. Please rephrase line 225 on page 11: it is not clear to which % they refer or maybe the authors forgot to report the number % of the second level of care. In Table 1 please correct "Age (SD)" moving (SD) after mean.

6) The discussion should be better rewritten. First, a summary of the main results: please first state the more important ones (related to the outcome) and then the secondary ones. Then, discuss each point (the reader would appreciate if subheading were used). The novelty of this study is not highlighted: the reader does not catch what this study adds to the existing literature. No mention is given about the % range of prescription of each antibiotic (nitrofurantoin, fluoroquinolones...) in the existing literature and in the international context. A range should be provided, since the prescription pattern is a main outcome and the reader should understand how much the % of prescriptions of fluoroquinolones (appropriate/not appropriate) found in the study, for example, differs compared to the range found in the literature.

Moreover, not being able to differentiate between complicated/uncomplicated should be stated among the limitations because this limits the interpretation of the results compared to the literature, since the authors referred to studies of appropriateness for uncomplicated UTI.

7) Minor: line 199 please correct "utilized" with "use". That's sounds better.

7. PLOS authors have the option to publish the peer review history of their article (what does this mean?). If published, this will include your full peer review and any attached files.

Reviewer #2: **Yes: **Nicholas A. Turner

Reviewer #3: **Yes: **Syed Muhammad Zaigham Abbas Naqvi

Reviewer #4: No

---

## [Author Response · Author response to Decision Letter 2]

18 Oct 2023

Point by point response letter.

Reviewer #2: 

“No addition comments from prior - I believe the authors have addressed all reviewers' points to the fullest extent possible”

We appreciate the feedback provided by the reviewer and are pleased to acknowledge that he believes the authors have adequately addressed all points raised. His input has been invaluable in enhancing the quality of our manuscript.

Reviewer #3: 

“The manuscript in current form should be acceptable in current shape. The manuscript in current shape is seemed to be properly formatted and incorporated with comments. There is no need for further changes, therefore I recommned this manuscript to be accepted by the editor.”

We express our sincere gratitude for the positive feedback of the reviewer on the current state of our manuscript. His thorough evaluation and endorsement for its acceptance by the editor are truly appreciated.

Reviewer #4: 

“1) Data availability issue: the data are made available by the authors but, in the way they are shared, they are not helpful to reproduce the main analysis. First, a code-book description and explanation of each variable (please rename all variables in English) with the specification of the coding used is missing and it seems that some variables are not provided. Please also specify if they refer to prescribers or patients.”

Thank you for your feedback and careful evaluation of our data availability. We sincerely appreciate your time and attention to detail in assessing our manuscript.

In response to your concern, we have provided a comprehensive data set in English, including the names and meanings of the variables, along with the corresponding codes for each variable used in the analysis. We have ensured that the data set Version 2 is structured to enable a clear understanding of each variable and its relevance to the study. We have also specified whether the variables refer to prescribers or patients, enhancing the clarity of our data set.

Please note that we have incorporated only the variables that are directly relevant to the study, ensuring that the provided data set aligns precisely with the scope of our research. The data is available in: https://data.mendeley.com/datasets/styystf2nj/2

We hope these modifications address your concerns adequately. Thank you once again for your valuable input; your feedback has significantly contributed to the enhancement of our work.

“2) In the methods, it should be clarified that the sample does not contain repeated measurements within patients. The main outcome and variables collected should be better specified and defined. In fact, it is not clear if the main outcome is the prescription pattern or the appropriateness of the prescription (I understand the second). This should be better highlighted and each variable listed specifying how it is defined (categorical, continuous ...). Otherwise, a more detailed data description should be provided as Supplemental. Please also refer the reader to the data shared (in a better understandable way).”

Thank you for your invaluable suggestions. We have duly clarified in the methods section (lines 167-168) that the sample excludes repeated measurements within patients. Regarding the main outcomes, we appreciate your insight, and in the methods section, specifically under the subheading "Design and Setting", in lines 124-126, indicates that our study aimed to analyze both antibiotic prescription patterns and the appropriateness of prescriptions concerning the clinical features of the patients.

Understanding the need for detailed variable descriptions, we have taken your suggestion seriously. We have included a comprehensive breakdown of each variable, specifying their definitions and categorizations, in Supplementary Material 1 (referenced in lines 190-191). Your meticulous review has significantly contributed to refining our manuscript, and we sincerely appreciate your valuable feedback.

“3) The statistical analysis does not take into account the correlation of patients followed by the same prescriber. In fact, it's not sufficient to adjust for prescribers' characteristics but also to adjust for the prescribers as random effects. That is, instead of a simple multivariable logistic regression, a mixed regression model should be performed. Moreover, the mean(sd) of the number of patients seen by the same prescriber is not reported. In the discussion, lines 303-304, page 15, it's reported that general practitioners and specialists were responsible for almost the same number of diagnoses of UTI but this is not evidenced in the results. Moreover, the authors wrote "bivariate analysis" but this is in fact "univariate regression analysis". Please correct it in line 192, methods and line 272, results”

Mixed-effects regression models are commonly used in longitudinal studies to analyze repeated data from the same subjects over time. These models are particularly suitable for handling the correlation between repeated observations of the same individual. Our study had a cross-sectional design, where observations were made at a single point in time, and subjects were not followed over time. Consequently, our sample does not contain repeated data for each individual. Since mixed-effects regression models are specifically designed for repeated data, their use in cross-sectional studies is neither common nor appropriate, as the data structure does not align with the nature of these models. Taking these considerations into account, we have decided to maintain the analysis through multivariable logistic regression. We hope the reviewer understands the rationale behind this decision.

Reporting mean and standard deviation of the number of patients seen by each prescriber is not pertinent to the outcomes we analyze. Our analysis aligns with methodologies employed in studies conducted across various contexts. Despite conducting a thorough search for additional studies to comprehend the possible significance of this result, we did not find any research that considered this specific aspect.

We have rephrased the sentence that may have been confusing, indicating that general practitioners and specialists were responsible for almost the same number of UTI diagnoses in lines 308-309.

We apologize for the confusion regarding the terminology used. We have corrected the terms and changed "bivariate analysis" to "univariate regression analysis" in lines 195 and 277. We appreciate the reviewer’s attention to detail regarding our study. 

“4) The authors have not appropriately addressed the previous reviewers' comment. If a table with the subgroup specification uncomplicated/complicated was not possible due to the lack of information, a subgroup analysis, based on the element/factors which defined appropriateness of prescriptions should be given. In fact, it's not clear how many ciprofloxacin prescriptions, for example, were appropriate or not.”

We appreciate the reviewer's attention to our study. Regarding the categorization of urinary tract infections as complicated or uncomplicated, we faced challenges due to numerous clinical factors (such as the lack of information about anatomical abnormalities of the urinary tract) and microbiological factors (like the presence of multi-resistant bacteria) that are essential for such classification. Given the complexity of these variables and the nature of our study design, precise categorization proved unfeasible. These limitations were duly acknowledged by previous reviewers and were understood, these constraints have been detailed in lines 427-436. Furthermore, performing a subgroup analysis would have similar limitations since this information is not available to ensure accurate classification, compromising the accuracy of our data.

Our methodology was specifically designed to assess the appropriateness of antibiotic prescriptions based on clinical features, as detailed in the "Assessment of appropriateness of antibiotic prescription" section of our methods. Unlike evaluating specific antibiotic choices, our focus was on the clinical conditions justifying a prescription, aligning with established clinical guidelines. These guidelines recommend first, and second-line antibiotics based on clinical features; hence, the decision to align with recommendations.

For example, the reviewer's mention of evaluating the appropriateness of ciprofloxacin (or any type of antibiotic) prescriptions underscores our point. In current medical practice, ciprofloxacin is no longer the recommended first-line treatment for cystitis due to evolving antibiotic resistance patterns. Consequently, any prescriptions of this antibiotic would be considered inappropriate. However, the choice of antibiotics relies on localized antibiotic resistance data, an aspect recommended in guidelines, a crucial aspect that we extensively address in our discussion, we contrast guidelines recommendations with our local patterns of antibiotic resistance. Finally, the factors influencing practitioners' antibiotic selection were not assessed and fall outside the scope of our research. 

We hope this clarifies our approach. Once again, we express our gratitude for the reviewer's thorough analysis, which has enhanced the depth of our study.

5) Results presentation is not systematic and clear. The authors should present more results in Tables and not in plain text, so that the reader could better follow and understand where reported results come from. In Table 1, please also report the count (%) of the patients with no comorbidities. The reader should catch immediately all relevant information. Please rephrase line 225 on page 11: it is not clear to which % they refer or maybe the authors forgot to report the number % of the second level of care. In Table 1 please correct "Age (SD)" moving (SD) after mean.

We sincerely appreciate the meticulous feedback provided by the reviewer. It's important to note that some of these observations had been previously raised and successfully resolved based on previous suggestions from other esteemed reviewers. We value this collective scrutiny, which significantly contributes to the enhancement of our manuscript.

In terms of the presentation of results, we aimed to strike a balance between tables and text, aiming for a comprehensive and reader-friendly approach. While certain data complexities necessitated textual explanations, we have diligently incorporated as much information as possible within tables to ensure clarity.

Regarding the concern about the percentage of UTIs in the second level, we have included this information in lines 209-201, precisely detailing its context for readers' understanding.

Moreover, we have thoroughly reviewed Table 1, implementing the necessary adjustments. The count and percentage of patients without comorbidities have been included to facilitate immediate comprehension. Additionally, we have restructured the presentation of age, placing (SD) after the mean, in accordance with the reviewer's recommendation.

We trust that these refinements have further polished the systematic and coherent presentation of our results, thanks to your invaluable guidance.

6) The discussion should be better rewritten. First, a summary of the main results: please first state the more important ones (related to the outcome) and then the secondary ones. Then, discuss each point (the reader would appreciate if subheading were used). The novelty of this study is not highlighted: the reader does not catch what this study adds to the existing literature. No mention is given about the % range of prescription of each antibiotic (nitrofurantoin, fluoroquinolones...) in the existing literature and in the international context. A range should be provided, since the prescription pattern is a main outcome and the reader should understand how much the % of prescriptions of fluoroquinolones (appropriate/not appropriate) found in the study, for example, differs compared to the range found in the literature. Moreover, not being able to differentiate between complicated/uncomplicated should be stated among the limitations because this limits the interpretation of the results compared to the literature, since the authors referred to studies of appropriateness for uncomplicated UTI.

We appreciate the reviewer's input. We have made modifications to the discussion section's structure, aiming to balance previous feedback from reviewers who endorsed the existing structure with the need to contextualize the results concerning the main outcomes: antibiotic prescription patterns and appropriateness. We hope the reviewer understands our reasoning; our intent was not to dismiss prior suggestions from other respected reviewers. Additionally, we have incorporated subheadings for better understanding, as the reviewer suggest keeping the guidelines submission of the journal.

In lines 354-362, we have incorporated the predominant antibiotic usage ranges in UTIs as suggested, providing readers with a reference point for our results. It's crucial to emphasize once more that our study evaluates the appropriateness of antibiotic prescriptions based on clinical features. We compare the use of first and second-line antibiotics for UTIs in alignment with guidelines and local antibiotic resistance patterns, taking the specific local context into account. This aspect represents the novelty of our approach—striving to contextualize clinical features within the specific setting where physicians practice. Furthermore, our study stands as one of the few reporting these outcomes in South America. 

In addressing the study's limitations related to the challenge of accurately distinguishing between complicated and uncomplicated UTIs, we elaborated on these aspects in lines 427-436."

We greatly value the constructive feedback provided by the reviewer, and we assure that all comments have been thoroughly considered to enhance the quality of our manuscript.

7) Minor: line 199 please correct "utilized" with "use". That's sounds better.

Thanks for the suggestion. We have changed the word utilized for used. Line: 203

---

## [Decision Letter · Decision Letter 3]

3 Nov 2023

PONE-D-23-12797R3Antibiotic prescription patterns in patients with suspected Urinary Tract Infections in EcuadorPLOS ONE

Dear Dr. Jimbo,

Thank you for submitting your manuscript to PLOS ONE. After careful consideration, we feel that it has merit but does not fully meet PLOS ONE’s publication criteria as it currently stands. Therefore, we invite you to submit a revised version of the manuscript that addresses the points raised during the review process.

We look forward to receiving your revised manuscript.

Kind regards,

Kwame Kumi Asare, Ph.D

Academic Editor

PLOS ONE

Reviewers' comments:

Reviewer's Responses to Questions

**Comments to the Author**

1. If the authors have adequately addressed your comments raised in a previous round of review and you feel that this manuscript is now acceptable for publication, you may indicate that here to bypass the “Comments to the Author” section, enter your conflict of interest statement in the “Confidential to Editor” section, and submit your "Accept" recommendation.

Reviewer #4: (No Response)

2. Is the manuscript technically sound, and do the data support the conclusions?

Reviewer #4: Yes

3. Has the statistical analysis been performed appropriately and rigorously? 

Reviewer #4: No

4. Have the authors made all data underlying the findings in their manuscript fully available?

Reviewer #4: Yes

5. Is the manuscript presented in an intelligible fashion and written in standard English?

Reviewer #4: Yes

6. Review Comments to the Author

Reviewer #4: The authors did not address one of my previous comments regarding the use of mixed models. The authors should perform a model with prescribers as random effects because in the study there are repeated measurements within prescribers, though it is true that in the study there are no repeated measurements within patients and the study is no longitudinal. However it is also true that in the study prescribers characteristics are included in the model (this is correct) but patients followed by the same prescriber are correlated and this should be accounted for in the adjusted model. Maybe the results would not change so much and the conclusions will remain the same. Anyway the authors should perform this analysis. The rest of the manuscript sounds good and once performed this analysis and correct the methods and results accordingly, the manuscript would be suitable for publication (unless other issues would not come out after this analysis).

7. PLOS authors have the option to publish the peer review history of their article (what does this mean?). If published, this will include your full peer review and any attached files.

Reviewer #4: No

---

## [Author Response · Author response to Decision Letter 3]

7 Nov 2023

Reviewer #4: 

“The authors did not address one of my previous comments regarding the use of mixed models. The authors should perform a model with prescribers as random effects because in the study there are repeated measurements within prescribers, though it is true that in the study there are no repeated measurements within patients and the study is no longitudinal. However it is also true that in the study prescribers characteristics are included in the model (this is correct) but patients followed by the same prescriber are correlated and this should be accounted for in the adjusted model. Maybe the results would not change so much and the conclusions will remain the same. Anyway the authors should perform this analysis. The rest of the manuscript sounds good and once performed this analysis and correct the methods and results accordingly, the manuscript would be suitable for publication (unless other issues would not come out after this analysis).”

Thank you for your insightful comments and recommendations. We acknowledge the importance of accounting for the correlation among patients followed by the same prescriber. In response to your suggestion, we have employed a mixed effects logistic regression model, incorporating a random effect for health professionals (prescribers), as detailed in the Methods section (lines 191-209). This adjustment has been meticulously applied in our updated analysis, which is now presented in Table 4.

Upon careful examination of the outcomes, we noted minor variations in some decimal estimations, with no substantial changes in the overall results. Crucially, our conclusions have remained consistent despite these adjustments. We have also updated the abstract to reflect these new findings, ensuring the manuscript's accuracy and completeness.

We genuinely appreciate your thoughtful critique, which has undoubtedly enhanced the robustness of our analysis and the final version of our manuscript.

---

## [Decision Letter · Decision Letter 4]

20 Nov 2023

Antibiotic prescription patterns in patients with suspected Urinary Tract Infections in Ecuador

PONE-D-23-12797R4

Dear Dr. Jimbo,

We’re pleased to inform you that your manuscript has been judged scientifically suitable for publication and will be formally accepted for publication once it meets all outstanding technical requirements.

Kind regards,

Kwame Kumi Asare, Ph.D

Academic Editor

PLOS ONE

Additional Editor Comments (optional):

Reviewers' comments:

Reviewer's Responses to Questions

**Comments to the Author**

1. If the authors have adequately addressed your comments raised in a previous round of review and you feel that this manuscript is now acceptable for publication, you may indicate that here to bypass the “Comments to the Author” section, enter your conflict of interest statement in the “Confidential to Editor” section, and submit your "Accept" recommendation.

Reviewer #4: All comments have been addressed

2. Is the manuscript technically sound, and do the data support the conclusions?

Reviewer #4: Yes

3. Has the statistical analysis been performed appropriately and rigorously? 

Reviewer #4: Yes

4. Have the authors made all data underlying the findings in their manuscript fully available?

Reviewer #4: Yes

5. Is the manuscript presented in an intelligible fashion and written in standard English?

Reviewer #4: Yes

6. Review Comments to the Author

Reviewer #4: I have no other comments. The authors addressed all my previous comments. The manuscript could be accepted.

7. PLOS authors have the option to publish the peer review history of their article (what does this mean?). If published, this will include your full peer review and any attached files.

Reviewer #4: No

---

## [Editor Report · Acceptance letter]

22 Nov 2023

PONE-D-23-12797R4 

Antibiotic prescription patterns in patients with suspected Urinary Tract Infections in Ecuador 

Dear Dr. Jimbo-Sotomayor:

I'm pleased to inform you that your manuscript has been deemed suitable for publication in PLOS ONE. Congratulations! Your manuscript is now with our production department. 

Kind regards, 

on behalf of

Dr. Kwame Kumi Asare 

Academic Editor

PLOS ONE